# ArtiSential^®^ Articulated Wrist-Like Instruments and Their First Application in Pediatric Minimally Invasive Surgery: Case Reports and Literature Review of the Most Commonly Available Robot-Inspired Devices

**DOI:** 10.3390/children8070603

**Published:** 2021-07-17

**Authors:** Giovanni Parente, Eduje Thomas, Sara Cravano, Marco Di Mitri, Marzia Vastano, Tommaso Gargano, Tosca Cerasoli, Francesca Ruspi, Michele Libri, Mario Lima

**Affiliations:** 1Pediatric Surgery Department, IRCCS Sant’Orsola-Malpighi University Hospital, 40138 Bologna, Italy; edu.thomas92@gmail.com (E.T.); sara-cravano@libero.it (S.C.); marcodimitri14@gmail.com (M.D.M.); marzia.vastano@icloud.com (M.V.); tommaso.gargano2@unibo.it (T.G.); toscacerasoli96@gmail.com (T.C.); francesca.ruspi@studio.unibo.it (F.R.); mlibri31@yahoo.it (M.L.); mario.lima@unibo.it (M.L.); 2Minimally Invasive and Robotic Pediatric Surgery Center (MISCBO), Alma Mater Studiorum—University of Bologna, 40138 Bologna, Italy

**Keywords:** minimally invasive surgery, robot-like instruments, wrist, wristed instruments, articulated instruments, surgical dexterity, laparoscopy, thoracoscopy

## Abstract

Background: Robotic surgery is currently a reality in surgical practice, and many endeavors have been made to extend its application also in pediatric surgery. In the absence of easy access to a robotic surgical system, new devices have been developed to offer a valid alternative such as wristed instruments. These differ from conventional laparoscopic instruments owing to a wrist-like mechanism at the tip, which faithfully reproduces the movements of the surgeon’s hands, regaining more movement’s degrees; Methods: We present two case reports in which the patients were subjected to minimally invasive procedures with aid of wristed instruments, followed by a review of the literature regarding the devices commonly marketed; Results: Articulated or wristed instruments render the same features seen in robotic surgery, such as major dexterity in smaller spaces, restitution of more natural movements and the ability to get over obstacles in a direct visual line. Our center recently equipped with ArtiSential^®^ articulated instruments and so far, they have proven to be of great value; Conclusions: wristed instruments could represent a standpoint for surgeons wanting to benefit from the advantages of robotic surgery with a cost-sensitive perspective.

## 1. Introduction

The robotic system had a great impact on the surgical community: 3D visualization, the ability to reduce tremors during movements, and the articulation of the instruments are the best known and more significant advantages.

However, an important drawback is the high costs that robotic surgery brings with it, which not all hospitals can afford. Furthermore, the instruments still must undergo a process of miniaturization before it can be stated that robotic surgery can become a routine in pediatric surgery.

Recent advances in technology have brought minimally invasive instruments inspired by robotic design onto the market to translate one of the most remarkable advantages of the robotic system to traditional minimally invasive surgery: these instruments are defined as “articulated” or “wristed”.

Our institution is equipped with one of them: ArtiSential^®^, an articulated device produced by Livsmed.

This paper aims to share our preliminary experience with articulated instruments describing two cases treated with ArtiSential^®^ and to present a review of the literature on the state of the art of the actual offer in the field of articulated instruments.

## 2. Case Reports

### 2.1. Thoracoscopic Thymectomy

#### 2.1.1. Case Presentation

A boy, aged 14 years old, was admitted to the Emergency Department in June 2020 complaining of fatigue, lower limbs hyposthenia, painful muscular cramps, encumbering deambulation, and difficulties getting up when sitting on the floor. Symptoms appeared nine months ago, when he re-started his usual physical activity, after a short break due to the COVID-19 pandemic. The physical exam revealed normal muscle tone and strength, early leg dropping during Mingazzini’s lower limbs test, pain during palpation of the right thigh, difficulties during heel-to-shin test, positive Babinski’s sign bilaterally, negative Romberg sign, steady gait with a normal base, intact coordination as measured by heel and toe walk. Laboratory and instrumental examinations were conducted, including biochemical markers of muscular damage, electroneuronography, and electromyography, and no relevant alteration was detected so the patient was discharged. After a month, the patient returned to the ER with a worsened clinical picture, complaining of dysphagia to both solids and liquids, dysarthria, speech difficulties, and a cutaneous rash on the lower limbs related to physical effort. The neurological profile was comparable to the previous one except for major pain intensity and hyposthenia in the upper limbs. During hospitalization, the patient’s symptoms deteriorated: his speech and gait became more impaired, he developed mild palpebral ptosis and an X-ray with contrast showed poor middle and inferior pharyngeal muscles contraction. Among other instrumental exams performed, ENoG and EMG of the right ulnar and facial nerves tested positive for a neuromuscular junction disorder; acetylcholine receptor (AChR) antibody test turned out positive, and the diagnosis of Myasthenia Gravis was confirmed. MRI images showed an enlarged thymus in the anterosuperior region of the mediastinum, with a retrosternal thickness of 2 cm, a transverse diameter of 7 cm, and a longitudinal diameter of 9.5 cm.

Therapy with pyridostigmine was started (15 mg × 3/day). During the subsequent follow-up (MG-ADL, MG-QoL15r, MGC, and QMG scores), due to the unsatisfactory response, doses were increased (up to 60 mg × 4/day) and prednisone was added (starting 25 mg/day, up to 50 mg/day) to the therapy. IVIG treatment (0.4 g/Kg/day) started a month and a half after the diagnosis and was associated with a slow and progressive remission of most of the symptoms.

#### 2.1.2. Surgical Treatment

Thoracoscopic thymectomy was performed under general anesthesia. The patient was intubated supine, and an endobronchial blocker was inserted into the main left bronchus to achieve single lung ventilation. The patient was then rolled into a right decubitus position to better expose the left hemithorax, with the left arm raised above his head and bent at a 90 degrees angle. Due to the narrow working space offered by the hemithorax, ArtiSential^®^ (LIVSMED, San Diego, CA, USA) articulated instruments were chosen to perform this procedure, to gain major dexterity by exploiting the wrist-like mechanism (Figure 1A–C). A 5 mm camera port was placed along the left midaxillary line at the fourth intercostal space, and under direct camera vision two 8 mm operating ports were placed respectively at the third intercostal space along the left posterior axillary line and the sixth intercostal space along the left midaxillary line. Lastly, a 5 mm port for a pulmonary retractor was located within the fifth intercostal space on the left posterior axillary line and was used to retract the collapsed lung posteroinferiorly, thus allowing one to identify the pericardial area, thymic bulge, and the left phrenic nerve posterior to the left internal mammary vessels. The mediastinal parietal pleura was opened with laparoscopic scissors above the thymic bulge, between the mammary vessels and the left phrenic nerve. Thymic tissue was identified alongside perithymic fat and with a laparoscopic grasper, they were gently grabbed addressing great attention not to exercise excessive traction on the tissues. Resection started from the left inferior horn and was performed using a blunt tip dissector to minimize the risk of iatrogenic damage. Thymus and mediastinal fat were mobilized and resected together to achieve maximum radicality. After the thymus was freed from the pericardium inferiorly, dissection continued along the anterior and posterior plane, lateral-to-medial across the midline, until the right pleura and phrenic nerve were identified. Thymic mobilization was completed by the dissection of its right horn and the division of the thyrothymic ligament. The specimen was removed in a plastic retrieval bag through one of the 8 mm ports. Ligation of significant blood vessels was not necessary, as the dissection was performed with a blunt dissector and hemostasis with an L-hook cautery and LigaSure^®^ (Medtronic Italia SpA, Milan, Italy). A chest tube was placed, and the lung was reinflated under direct camera vision. All port sites were closed in layers using adsorbable sutures. The entire operation was performed without complications and with minimal bleeding.

### 2.2. Hepatic Lymphangioma Debulking

#### 2.2.1. Case Presentation

A 9-year-old boy was admitted to our emergency department with abdominal pain and dyspnea. At physical examination, he presented with abdominal pain at palpation in the periumbilical and right lower quadrant, rebound tenderness, and a positive McBurney sign. Abdominal ultrasound performed with the suspicion of acute appendicitis, instead revealed hyperechogenicity of the tissues surrounding the portal vein, gallbladder, duodenum, stomach, and dorsal pancreatic surface, with various fluid filled cysts included. Therefore, the patient was admitted for further investigations, and he was subjected to an MRI scan, which revealed the presence of an intra-hepatic lymphangioma extending outside the liver and infiltrating the gallbladder and pancreas. A laparoscopic biopsy was performed, and the histology report confirmed the diagnosis of lymphangioma.

#### 2.2.2. Surgical Treatment

The laparoscopic approach under general anesthesia was the method of choice. Due to the proximity to the extra-hepatic biliary tree, we administered indocyanine green to the patient the night before the procedure, thus making the biliary structures recognizable thanks to the fluorescence detected by the IMAGE 1™ S RUBINA™ camera (Karl Storz Endoscopia Italia S.r.l., Verona, Italy). In addition, the restricted working space under the liver led us to choose ArtiSential^®^ wristed instruments to achieve greater maneuverability and dexterity (Figure 1D–E). A 10 mm camera port was placed at the umbilicus. A 5 mm port was placed in the left upper quadrant for the liver retractor. Two 8 mm ports were respectively positioned in right and the left flank for the articulated instruments. Careful dissection was carried out along the planes separating the duodenum and the gallbladder from the lymphangioma, using blunt laparoscopic swabs and the wristed monopolar hook. Final dissection of the exophytic intraperitoneal lymph-angiomatous tissue was carried out with LigaSure. An abdominal Jackson-Pratt abdominal drain was left in place.

## 3. Discussion

The robotic system, without doubt, has several advantages that make it popular for use in surgery. Unfortunately, high costs limit its use making it available only in affluent settings; moreover, in pediatric surgery, the need for smaller dimension instruments determines the applicability of robotic surgery only to very selected cases.

Biomedical engineers, to overcome the above disadvantages, developed laparoscopic instruments with new features that could both resemble robotic surgery in terms of motion dexterity enhancement and answer the need for affordability and computer or motor absence: these instruments are defined as “wristed” or “articulated”.

Conventional laparoscopic instruments are structured as shown in Figure 2A: they consist of a handle (which is the surgeon’s interface of the device) and a shaft that, in pediatric surgery, is available in different lengths and diameters according to the soma of the patients, not deflectable and that ends with the tip, the feature of which characterizes the instruments itself (a grasp, a forceps, a needle-holder and so on). The opening and closure of the handle determine, respectively, the opening and closure of the jaws at the tip of the instruments. As the instruments are inserted into the body of the patient through a trocar, it creates a fulcrum effect that limits movement to 4 degrees of freedom (as shown in Figure 2B), which can become 5 if we consider the open/close motion of the jaws at the tip.

Presented briefly are the basis of the structure of a conventional laparoscopic instrument, it is possible now to go through the articulated devices proposed in the literature.

Wristed instruments can be classified based on the subsequent characteristics:(a)Type of wrist-like mechanism(b)Type of movements control at the handle(c)Direction of wrist movement compared to one imparted to the handle

The wrist-like mechanism realized essentially in two different manners, divides the instruments into two categories: ones that have zero ben radius when deflected and the other that, when deflected, bend into a curved arc (Figure 2C). Generally, in the former, the jaws are included in the wrist mechanism itself while in the latter the jaws are located distal to the wrist.

As regards to most of the types of movement control, we can distinguish a handle control, a thumb control, or a mixed one: the handle control enables the handle itself to control the wrist movements, the thumb control method consists of a handle rigidly attached to the shaft and the surgeon can control the wrist via a joystick or a ball manipulated by the user’s thumb; finally, the mixed one’s wrist degree of freedom is controlled with a different knob or lever on the handle.

Point (c) refers to the direction of wrist movement when mapped to the user control interface: Anderson P.L. classified it in “parallel”, when the mapping of the control handle to the end effector is such that the central axes of the two remain approximately parallel during articulation, and, if not, “reverse” [1].

Some of the most popular articulated tools discussed in literature will be presented below.

### 3.1. Radius Surgical System^®^ (Tuebingen Scientific Medical GmbH, Tübingen, Germany)

The Radius Surgical System^®^ (RSS, Figure 3A) consists of two hand-guided surgical manipulators and provides a deflectable and rotatable tip allowing six degrees of freedom. The wrist deflects unidirectionally with respect to the shaft of the device and axial rotation of the shaft is achieved via a thumb knob. A large, multi-finger trigger on the handle operates the jaws at the end effector which cannot be locked. The diameter of the shaft is 10 mm.

The re-usable Radius Surgical System^®^ (RSS) is equipped with disposable different effectors that can easily be fixed to the tip of the instrument depending on the surgical task required; actually, the effectors produced are different types of needle-holders, an atraumatic grasping forceps and a gastric band retractor [2,3,4].

### 3.2. FlexDex^®^ (FlexDex Surgical Inc., Brighton, MI, USA)

This device has a quite unique user interface in which the shaft of the instruments is connected to the user’s forearm via a bracelet linked to the frame of the tool, thus the structure described places the wrist articulation axes at the center of the surgeon’s wrist (Figure 3B) [5,6,7,8,9].

The surgeon’s wrist deflections are mapped to wrist effector deflections inside the patient by two strips determining a parallel mapping.

The FlexDex enables wrist degrees of freedom to be decoupled from the conventional four degrees of freedom of standard laparoscopic instruments. The surgeon controls the standard four degrees of freedom with his or her forearm, and the wrist deflection with his or her wrist. The shaft diameter is 8 mm.

The concept of a 1:1 mapping of the surgeon’s wrist to instrument wrist inside the patient make this instrument appealing but two principal drawbacks are to be noticed: the instrument is equipped with just a needle-holder and the forearm attachment makes impossible a quick detachment and substitution of the instrument in case of emergency.

### 3.3. LaproFlex^®^ (DEAM, Roden, The Netherlands)

The LaproFlex^®^ instrument (Figure 3C) consists of a handle that, deflected by the surgeon’s wrist, causes the tip of the instrument to bend in a reverse fashion when mapped to wrist motions. The handle has a lever on which the operator, by exerting pressure with the second and third fingers, determines the closure of the jaws. Furthermore, between the handle and the shaft of the instrument, there is a knob which, rotating, turns the shaft by 360°. The shaft has a diameter of 5 mm [10].

LaproFlex^®^ has different models that change based on the end effector: a dissector, different kinds of graspers, dissectors and scissors, and a hook are available.

### 3.4. Intuitool^®^ (UNeMed Corporation, Omaha, NE, USA)

The Intuitool^®^ instrument (Figure 3D) consists of a pistol-handle grasp in which the handle has a trackball that can be rotated in the four directions of the space deflecting in the same way (a parallel mechanism) the tip of the instrument. The end effector jaws are operated by a trigger in the handle, and they can be locked with a button on the handle too. The shaft diameter is 10 mm [11,12].

### 3.5. ArtiSential^®^ (LIVSMED, San Diego, CA, USA)

The ArtiSential^®^ (Figure 3E) is an 8 mm diameter pistol-handle instrument. It has zero ben radius when deflected; its wrist function is guaranteed by two joints at the end effector that couples their movements with the ones of the handle [13,14,15,16,17,18].

Indeed, the handgrip can move on both the vertical and horizontal axis transmitting its movement to the end effector. The handle is equipped with jaw control rings with which the surgeon, using the thumb and the second finger, can open and close the jaws. The end effector can also be locked via a locking lever located on the handle too.

ArtiSential^®^ is single-use and is available in different forms differing from the type of the end effector: needle-holder, dissectors, clip applier, monopolar hook, and spatula.

All the previously reported instruments answer the need for robotic-like dexterity without significant robotic surgery costs.

Currently, even more devices are under development and refinement making this field a thriving scientific and economic one.

It is difficult to determine which of the previously mentioned instruments possesses the best characteristics because, in Literature, it is still debated.

As regards to the direction of kinematic mapping, discussed in point (b) of this paragraph, most of the devices show a reverse kinematic mapping; however, some authors criticize this type of mechanism as counterintuitive. Actually, in deep narrow spaces (such as a pediatric thorax), where the instruments’ shafts are near parallel and close to one another, a parallel mapping reduces the risk of surgeon’s hands colliding; on the contrary, if a wide triangulation is guaranteed, the reverse mapping will place surgeon’s hands in a comfortable location. Finally, it has to be considered that the more appropriate kinematic mapping could be a matter of the surgeon’s preference.

Even the design of the wrist is a matter of discussion. Indeed, some prefer a curved wrist since it resembles the tip of endoscopic instruments and, therefore, is something we are already used to. By the way, curved instruments come with an important drawback: their use required a large volume of space inside the patients especially ones with large radius curvature, and it is not always available in pediatric surgery. This disadvantage is overcome by pin joints that can also integrate the jaws to the joints themself reducing, even more, the workspace required for their utilization [19].

Even the better structural way to give the instruments axial shift rotation is discussed. Most of the presented devices have a dedicated knob in various positions to rotate the shift while some others required the surgeon to rotate the knob itself; we think the first mechanism is the preferable one.

The last point of discussion is the need to include in the anatomy of the devices a tool to be able to lock them. Jaw closure locks are a standard in laparoscopic instruments; therefore, we think it could be helpful to add them to wristed devices not to lose instrument functions but just add the novelty coming from the wrist.

ArtiSential^®^ articulated instruments have been recently purchased at our Surgical Unit. Since their implementation in our activity, they have proven to be invaluable when it comes to operating in narrow spaces and overcoming obstacles in the direct visual line of the surgeon. These are renowned benefits of the available surgical systems, which have now been transferred to the setting of a classic thoracoscopy or laparoscopy. As our surgical team is very skilled in laparoscopic procedures, the learning curve for this new equipment was quite steep.

The absence of a comparative study, preferably structured as a prospective randomized case-control trial, between conventional MIS/robotic surgery and MIS using wristed devices represents a limit of the present paper preventing us from asserting with certainty the superiority or at least non-inferiority of the described instruments vs. traditional ones. Nevertheless, this is not the porpoise of the present paper which is designed to share our preliminary experience and first impressions as long as to make pediatric surgeons aware of this novel technology.

We, therefore, encourage multicenter randomized case-control studies to endorse the adoption of these robot-inspired devices with stronger scientific evidence.

## 4. Conclusions

We have presented a vast array of articulated instruments currently available on the market. Their use in the pediatric field is still restricted due to the same limitations depicted for robotic surgical systems, such as the need for smaller instruments and the relatively high costs.

We have provided our preliminary experience on how these instruments can be safely used in a pediatric setting with great benefit to the surgeon and the patient, based on which we enthusiastically endorse multicenter randomized case-control studies to spread their application in MIS with stronger scientific evidence.

## Figures and Tables

**Figure 1 children-08-00603-f001:**
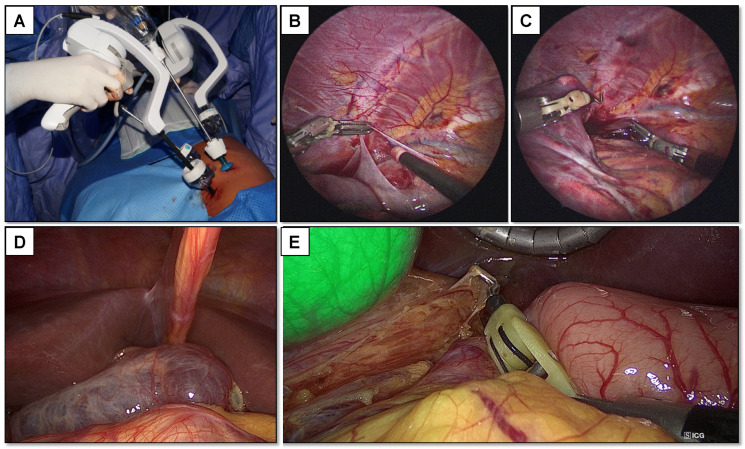
Thoracoscopic thymectomy performed using ArtiSential^®^ (**A**): some pictures taken from the surgical procedure in which the enhanced movements’ degrees of freedom even in a small operative space such as a pediatric thorax is clear (**B**,**C**). In (**D**,**E**) we can see the lymphangioma, resected using the wristed instrument with the aid of indocyanine green highlighting the biliary tree to avoid iatrogenic injury.

**Figure 2 children-08-00603-f002:**
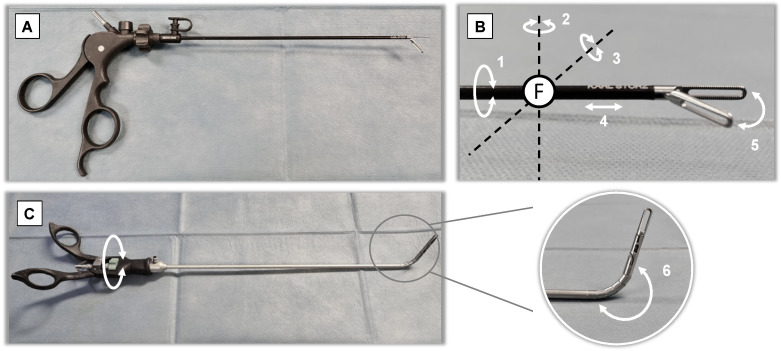
(**A**) Conventional laparoscopic instrument; (**B**) 5 movement’s degrees of freedom of a conventional laparoscopic instrument; (**C**) 6th degree of freedom given from a curved instrument. F = fulcrum (fulcrum effect given by trocars).

**Figure 3 children-08-00603-f003:**
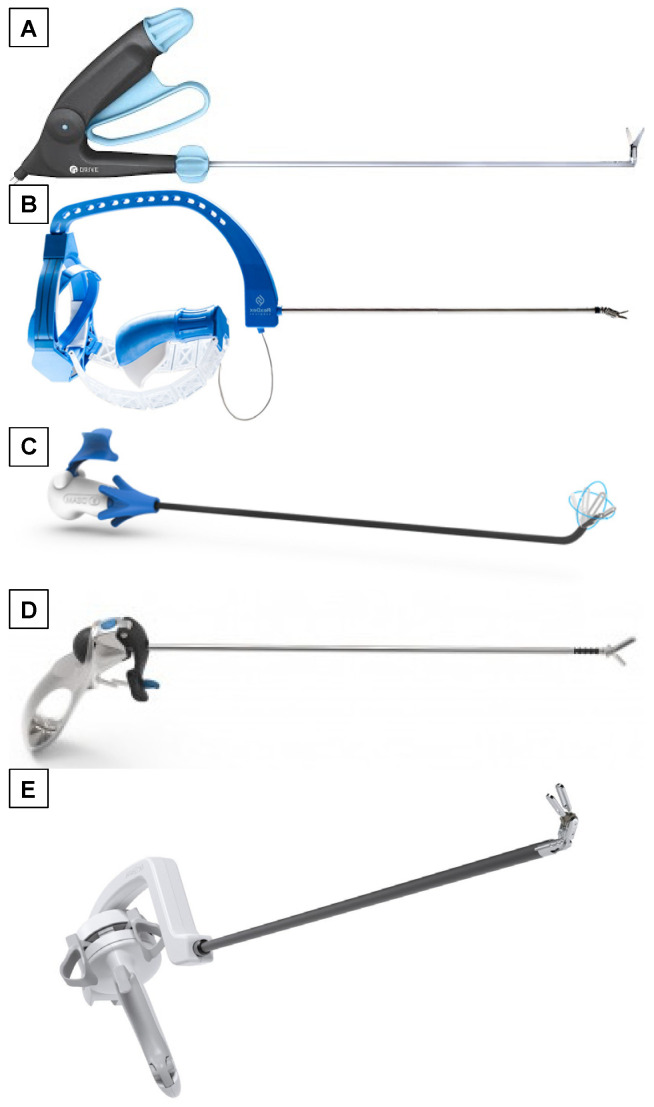
Most commonly available and updated wrist-like devices. (**A**) Radius Surgical System^®^ (Tuebingen Scientific), (**B**) FlexDex^®^ (FlexDex Surgical Inc.), (**C**) LaproFlex^®^ (DEAM), (**D**) Intuitool^®^ (UneMed), (**E**) ArtiSential^®^ (LIVSMED).

## Data Availability

The data presented in this study are available on request from the corresponding author. The data are not publicly available due to privacy restrictions.

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
