# Peer review of "ArtiSential® Articulated Wrist-Like Instruments and Their First Application in Pediatric Minimally Invasive Surgery: Case Reports and Literature Review of the Most Commonly Available Robot-Inspired Devices"

_children, 2021, doi:10.3390/children8070603_

Round 1
Reviewer 1 Report
This is an avarage job overall, since the report adds nothing new to all that can be found in the literature.
On the other hand both case reports regarding quite rare entities in pediatric patients are a very interesting read.
In fact, if you mention the benefits of articulated instruments for laparoscopic procedures and claim to have provided evidence for their safe use in pediatric patients a higher case count and use in children of younger age with smaller body cavities would improve the proposition already made here. There's also no direct comparison to robotic-assisted procedures or, if aiming at minimal invasiveness in the child, to less invasive and already established laparoscopic techniques such as single incision, e.g. in combination with wristed instruments.
Reviewer 2 Report
In this paper, the authors focused on the challenges for pediatric surgery as various techniques for minimally invasive surgery have been developing. Although the number of instruments that can be used in pediatric surgery is limited, the authors mentioned some devices inspired by robotic technology had been developed. In particular, the authors reported two surgical cases with ArtiSential(r) and introduced four related devices. And the patient information and treatment methods were carefully described as for the case report with ArtiSential(r). The following are my detailed comments:
(1) About the paper title
I think that this review paper is designed to introduce ArtiSential(r). For this reason, the title of the paper is uncomfortable for readers. In the paper, it was explained that it reviewed "articulated" instruments, but the majority of the content was about the cases in which ArtiSential(r) was used, rather than a review in which multiple instruments were compared fairly. In this sense, I think you should change the title more specific.
If you want to make this a review paper with this title, you should introduce the four instruments (you showed in Chapter 3) first, and then report the case reviews with ArtiSential(r). And also, you should describe clearly the reason you chose the instrument as a case review. I think the reason based on using this device in your institution was insufficient for using this title.
(2) About introducing related instruments
As a literature review, the introduction of the instruments in Chapter 3, most of the contents described ArtiSential(r), and there are very few reviews of other related instruments. If you want this paper to be a review paper on articulated instruments, you should devote the same amount of writing space to describe the features of each instrument, such as advantages and disadvantages of their use, examples of their use in pediatric surgery, etc. using references, so that readers can compare the instruments. In particular, you illustrated the usability of ArtiSential(r) only. But you did not compare with the use of other techniques. If you want to show this in this section, you should describe the feature of other instruments like the feature of ArtiSential(r). If it is difficult to describe it, the contents should be moved in the case report section.
(3) About figure
Figure 3 is not used in the paper. If this figure needs to use in the paper, you should show it by referring. I think that this figure shows one of the scenes in the case reports. If so, you should refer to it in chapter 2.
(4) Minor suggestion
The title of section 2.1 has a typo. thimectomy : thymectomy
Round 2
Reviewer 1 Report
Dear Colleagues,
I'm happy to contribute and be of some help during the reviewing process of your presented work. I'm glad that you considered the points mentioned in the 1st review and applied some significant changes.
Keep up the good work and good luck.
Reviewer 2 Report
I confirmed your revised manuscript.